# To Ask or Not To Ask: Robot-assisted Bite Acquisition with Human-in-the-loop Contextual Bandits

**Rohan Banerjee, Sarah Dean, Tapomayukh Bhattacharjee**
Department of Computer Science, Cornell University
{rbb242, sdean, tapomayukh}@cornell.edu

**Abstract:** Robot-assisted bite acquisition involves picking up food items that vary considerably in their shape, size, texture, and compliance. An effective bite acquisition system should be able to generalize to out-of-distribution instances in a few-shot manner, but this is difficult for a fully autonomous strategy due to the large variety of food items that exist. In this work, we propose a contextual bandit algorithm that asks for human feedback to improve generalization to novel food items, while minimizing the cognitive workload that querying imposes on the human. We demonstrate through experimentation on a dataset of 16 food items that our algorithm improves the tradeoff between task performance and cognitive workload compared to two baselines: (1) a state-of-the-art fully autonomous baseline, and (2) a naive querying algorithm that does not incorporate cognitive workload.

**Keywords:** Contextual Bandits, Bite Acquisition, Human-in-the-Loop Autonomy

## 1 Introduction

Eating is a ubiquitous activity of daily living [1], but approximately 1 million people in the U.S. are unable to eat without assistance [2]. Robot-assisted feeding could help address this gap, as a robotic system could feed a user with mobility limitations who cannot feed themselves independently. Robot-assisted feeding consists of two major components: bite acquisition [3–6], which is the task of picking up a food item, and bite transfer [7, 8], which is the task of transferring the food item to the mouth of the user. In our work, we focus on bite acquisition, with the goal of learning a policy that can robustly acquire food items with varying shape, color, texture, compliance, and size.

A typical grocery store contains more than 40,000 unique food items [9], motivating the need for robotic bite acquisition strategies that can generalize well to out-of-distribution food instances. Existing state-of-the-art approaches to robotic bite acquisition [4, 5] perform well when exposed to out-of-distribution food items, but require a large number of samples to converge to the optimal acquisition action. Our belief is that it would be very challenging for a fully autonomous strategy to achieve few-shot convergence to novel food items due to the significant diversity of food items. Unlike in the rigid body manipulation setting, food items are fragile, meaning that an inefficient acquisition strategy could destroy the food items.

To overcome this limitation, our insight is that we can leverage the presence of the human user to develop online strategies that query that user for feedback on novel food items. A human being could provide varying levels of feedback to the agent, such as identifying the food item or indicating where the food item is located. In our work, we assume that the human can give expert feedback, indicating what action the robot should take for the food item. However, frequent querying will impose a cognitive workload on the human, which will ultimately decrease technology acceptance.

Therefore, the key research question that we address in this work is as follows: How can we balance this trade-off between querying and task performance? In the setting where we receive expert labels from the human, we hypothesize that an algorithm that decides when to query based on the estimated cognitive workload penalty of querying the human, as well as the current uncertainty about the performance of different bite acquisition actions, will achieve higher task performance (convergence

7th Conference on Robot Learning (CoRL 2023), Atlanta, USA.

to the most performant acquisition) compared to a fully autonomous baseline, while achieving lower cognitive workload compared to a naive querying baseline that does not account for the workload.

Prior work has studied the problem of deferring to an expert [10–14], and prior work in shared autonomy for human-robot interaction has explored incorporating human state information into the robot's decision-making [15, 16]. Ours is the first to study deferring to a human in an online contextual bandit setting for the domain of robot-assisted bite acquisition, in which we incorporate a model of cognitive workload. We provide further discussion of related work in Appendix A.

Our contributions are as follows:

- We propose an extension to the contextual bandit framework to incorporate the joint problem of when to query and which robot action to select for bite acquisition, where we incorporate the cognitive workload of querying the human into the reward function.
- We develop two querying algorithms (one naive algorithm, and one informed algorithm that takes cognitive workload into account). We demonstrate through experimentation with a labeled set of real-world food images that the informed algorithm achieves a higher weighted combination of task reward and cognitive workload compared to a state-of-the-art strong LinUCB baseline.

## 2  Problem Formulation

We use an online learning formulation based on the contextual bandit approach used in prior bite acquisition work [4, 5]. In this online setting, we receive a context (corresponding to a food item) at each timestep, and we then must decide whether to query or to select one of the robot actions to pick up the food item. More formally, our learning agent receives a sequence of contexts $x \in \mathcal{X}$ (with no distributional assumptions on $x$), and must select actions $a \in \mathcal{A}$ that maximize expected reward. The two primary components of contextual bandit approaches are policy updating (how to update the policy based on reward observations) and action selection (how to select actions during the learning process) [17]. Our primary focus is on action selection, which we extend to consider the special query action. We develop our algorithm for a parameterized reward model learned by regression, but remark that our algorithms would work more generally for any optimism-based contextual bandit sub-routine.

Instead of explicitly detecting out-of-distribution contexts, we use the natural capability of a class of contextual bandit algorithms to implicitly detect distributional shifts by modeling parameter uncertainty. As we acquire data online, our algorithms construct reward confidence intervals that reflect the similarity between an observed context and previously seen contexts. In our problem setting, we assume that we have access to a dataset $D$ collected from a robotic manipulator, consisting of raw observations $o \in \mathcal{O}$, actions $a \in \mathcal{A}$, and rewards. We use $D$ to pretrain and validate our contextual bandit algorithms, but our algorithms also apply to the fully online setting without *a priori* access to $D$. The dataset $D$ and contextual bandit setting are characterized as follows:

- **Observation space** $\mathcal{O}$: RGB-D images that contain a single bite-sized food item belonging to a set of 16 distinct food types, situated on a plate under 3 distinct environment configurations: isolated, close to the plate wall, and stacked on top of another food item [3].
- **Action space** $\mathcal{A}$: A discrete action space shown in Figure 1 (left) consisting of 7 actions in total - 6 robot actions $a_r$ for $r \in \{1, \ldots, 6\}$ and 1 query action $a_q$. Each robot action is a pair consisting of one of three pitch configurations (tilted angled (TA), vertical skewer (VS), tilted vertical (TV)) and one of two roll configurations ($0°, 90°$), relative to the orientation of the food.
- **Context space** $\mathcal{X}$: As in [4], we derive a lower-dimensional context $x \in \mathbb{R}^{2048}$ from the visual observations $o$ using the SPANet network [3]. SPANet is pretrained in a fully-supervised manner to predict $s_o(o, a)$, the probability that action $a$ succeeds for observation $o$. A critical assumption for our bandit algorithms is the realizability of a linear relationship between $x$ and $r$. Because the final layer of SPANet is linear, we use the penultimate activations of SPANet as our context.

We model cognitive workload as one component of a time-dependent reward function $r(x, a, t)$, which returns the expected reward at time $t$ given context $x$ and action $a$. In our setting, when we query the human, we receive expert feedback in the form of the ground-truth action $a^*(x)$ that has

the highest probability of successfully picking up the food item contained in $x$, and this feedback incurs a cognitive workload penalty $c_{CL}(t) \geq 0$. We thus model our reward function as consisting of a task reward $r_{task}(x, a, t)$ and a cognitive workload reward $r_{CL}(x, a, t)$:

$$r_{task}(x, a, t) = \begin{cases} s_x(x, a) & a \neq a_q \\ s_x(x, a^*(x)) & a = a_q \end{cases} \qquad r_{CL}(x, a, t) = \begin{cases} 0 & a \neq a_q \\ -c_{CL}(t) & a = a_q \end{cases}$$

where $s_x(x, a)$ is the probability that action $a$ succeeds for context $x$. In the above expression, the task reward can be understood as the mean of a Bernoulli random variable. Note that the dataset $D$ contains observed binary rewards that can be thought of as samples from the Bernoulli distribution.

We define $c_{CL}(t)$ using a rebounding satiation model for multi-armed bandits [18] adapted to our contextual bandit setting. First, we define $v_t$ to be the cognitive workload state at time $t$, with dynamics $v_t = \gamma v_{t-1} + \gamma_q \mathbb{1}(a_t = a_q)$. In this model, $\gamma \in (0, 1]$ is the retention factor, $\gamma_q \in (0, 1]$ is the instantaneous cost, $a_t$ is the action taken at time $t$, and $\mathbb{1}(a_t = a_q)$ is equal to 1 if $a_t = a_q$ and 0 otherwise. We then define $c_{CL}(t) = \lambda_q v_t$, where $\lambda_q \geq 0$ is a scaling factor.

The cognitive workload penalty is higher when many queries have been made recently, while the impact of past queries is discounted over time. Our goal is to learn a policy $\pi(a|x, t)$ that maximizes the expected reward $\mathbb{E}[r_t|x_t, a_t, t] = r(x, a, t) = w_{task} r_{task}(x, a, t) + w_{CL} r_{CL}(x, a, t)$ over any observed sequence of food items, where $w_{task}$ and $w_{CL}$ are positive, scalar weights that sum to 1.

## 3  Querying Algorithms

Our fully autonomous baseline is LinUCB [19], which is the state-of-the-art algorithm for bite acquisition [4, 5]. LinUCB selects the action for an observed context $x$ that maximizes an upper-confidence bound (UCB) estimate of the reward for each possible action, given by $UCB_a = \theta_a^T x + \alpha b_a$. Here, $\theta_a$ is the parameter vector for a linear reward model that is learned through regression on a context matrix $X_a$ and observed rewards for each action $a$, where $X_a$ includes the contexts seen for action $a$ during pretraining and online validation, $\alpha > 0$ is a parameter that corresponds to a particular confidence level in the accuracy of the UCB estimate, and $b_a = (x^T (X_a^T X_a + \lambda I)^{-1} x)^{1/2}$ is the UCB bonus with $L_2$ regularization given by parameter $\lambda$. The size of the reward confidence interval $\alpha b_a$ reflects how out-of-distribution $x$ is compared to the contexts observed in $X_a$.

We consider the following two querying algorithms, where both algorithms decide to either query the human or select the robot action that maximizes $UCB_a$. Our first algorithm is called Exp-Decay and is shown in Algorithm 1. The Exp-Decay algorithm queries with an exponentially-decaying probability that is a function of the number of food items seen within an episode, and is independent of the query cognitive workload. Our second algorithm is called LinUCB-Query-Gap (abbreviated LinUCB-QG) and is shown in Algorithm 2. LinUCB-QG decides to query if the worst-case performance gap between the best action $a^*$ and second-best action $a^-$ exceeds a scaled version of $c_{CL}(t)$, with scaling factor $w$. A larger gap represents a greater risk that the predicted best arm may be suboptimal (for instance, if $x$ is out-of-distribution), increasing the odds that the reward gain from querying will exceed the cognitive workload penalty.

---
**Algorithm 1** Exp-Decay.

---
1: Inputs: Context $x$, decay rate $c$, number of food items seen $N$, time $t$
2: For all robot actions $a_r$, compute UCB bonus $b_{a_r}$ and UCB value $UCB_{a_r}$.
3: Set $P(query) = e^{-cN}$ if $t$ is the first timestep to observe $x$, $P(query) = 0$ otherwise.
4: Set $a = a_q$ with probability $P(query)$, $a = \arg\max_{a_r} UCB_{a_r}$ with probability $1 - P(query)$.
5: **return** $a$

---

## 4  Evaluation + Results

**Dataset.** We use the food dataset from [3], consisting of food images that were collected using a Kinova Gen-2 robot with a fork attached to its end effector. The robot arm was instrumented with an RGB-D camera on the end effector. In front of the robot arm was a plate containing bite-sized food

**Algorithm 2** LinUCB-QG.

1: Inputs: Context $x$, weight term $w$, time $t$
2: For all robot actions $a_r$, compute UCB bonus $b_{a_r}$ and UCB value $UCB_{a_r}$.
3: Let $a^* = \arg\max_{a_r} \theta_{a_r}^T x$, $a^- = \arg\max_{a_r \neq a^*} \theta_{a_r}^T x$
4: Set $a = a_q$ if $(\theta_{a^-}^T x + \alpha b_{a^-}) - (\theta_{a^*}^T x - \alpha b_{a^*}) > w c_{CL}(t)$, $a = \arg\max_{a_r} UCB_{a_r}$ otherwise
5: **return** $a$

items. The dataset contains 16 food types, with each food type including 30 trials for each of the 6 robot actions in the action space. Figure 1 (right) contains examples of food items from the dataset.

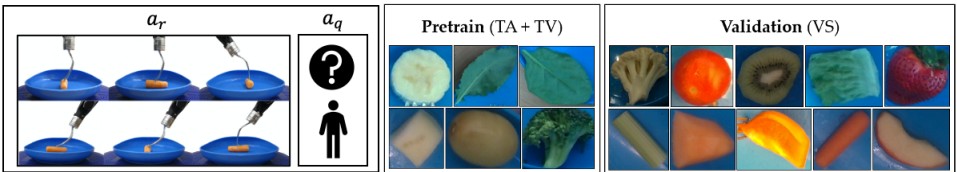

Figure 1: Action space and food items. *Left:* Image [3] of robot actions $a_r$, consisting of three distinct pitch configurations: tilted angled (TA) in lower left, vertical skewer (VS) in upper left, and tilted skewer (TV) in right, each with $0°$ and $90°$ roll configurations, along with query action $a_q$. *Right:* Food items and optimal pitch configurations for pretraining set $D_p$ and validation set $D_v$.

**Generalization scenarios and metrics.** To assess the generalization performance of our algorithms, we pretrain the LinUCB reward model with a subset of food items. We partition $D$ into a pretraining set $D_p$ and a validation set $D_v$, from which food instances are drawn online. We construct $D_p$ and $D_v$ so that the optimal actions for food items in each set are disjoint. In our case, $D_p$ includes 6 food types (optimal actions: TA or TV), while $D_v$ includes the remaining 10 food types (optimal action: VS), shown in Figure 1 (right). We measure the following metrics: the total task reward $r_{task}(x, a, t)$, the total cognitive workload reward $r_{CL}(x, a, t)$, a weighted metric $M_{wt} = w_{task} r_{task}(x, a, t) + w_{CL} r_{CL}(x, a, t)$, where $w_{task}$ and $w_{CL}$ represent user-specific preferences for the task reward/cognitive load tradeoff.

**Querying algorithm hyperparameters.** For the LinUCB baseline, we conduct hyperparameter tuning (see Appendix B) to select the value of $\alpha = 0.1$. For Exp-Query, we consider 4 different values of the decay rate $c$. For LinUCB-QG, we consider 6 different values of the weight term $w$. All experiments use the following cognitive workload parameters: $\lambda_q = 1$, $\gamma = 0.5$, $\gamma_q = 10^{-2}$.

## 5  Discussion

Table 1 shows results for a user preference of $w_{task} = 0.7$ and $w_{CL} = 0.3$ (see Appendix C for additional preference settings). LinUCB-QG achieves a higher weighted score compared to LinUCB (fully autonomous baseline) and Exp-Query (naive baseline) because in the above setting where the cost of querying is low, LinUCB-QG trades off task performance and cognitive workload by frequently querying (see Appendix C for additional workload settings). Figure 2 visualizes the metrics for each hyperparameter setting, showing that LinUCB-QG can manage user-specific tradeoffs.

| Method | $r_{task}$ | $r_{CL}$ | $M_{wt}$ |
|---|---|---|---|
| LinUCB | 0.716 | - | 0.501 |
| Exp-Query | 0.738 | **−0.026** | 0.509 |
| LinUCB-QG | **0.819** | −0.139 | **0.532** |

Table 1: Querying bandit algorithm metrics. Averages are across 5 random seeds (For Exp-Query, we also average across 5 policy random seeds). Values for $r_{task}$ and $r_{CL}$ correspond to hyperparameter setting with maximal $M_{wt}$.

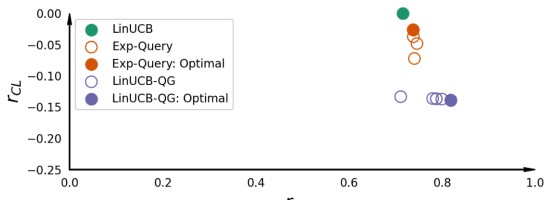

Figure 2: Pareto plot with $r_{task}$ and $r_{CL}$ for LinUCB and different settings of Exp-Query (varying $c$) and LinUCB-QG (varying $w$). Filled circles indicate the setting that maximizes $M_{wt}$.

**Acknowledgments**

This work was partly funded by NSF CCF 2312774 and NSF OAC-2311521, a LinkedIn Research Award, and a gift from Wayfair, and by NSF IIS 2132846 and CAREER 2238792. The authors would like to thank Ethan Gordon for his assistance with the food dataset.

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

# A Related Work

**Learning to defer.** Our problem setting is an instance of learning to defer to an expert [10]. In this setting, a learning agent must decide whether to produce decisions according to its own learned model, or defer to the decisions from one or more experts, such as oracle models or humans. Most approaches operate in the supervised learning [10–13] or RL [14] domains. Motivated by the constraints of the robot-assisted bite acquisition task, in which we observe independent contexts and receive sparse feedback in response to robot acquisition actions based on those contexts, our work focuses on developing learning-to-defer policies in an online contextual bandit setting. Some learning-to-defer approaches incorporate a fixed cost of querying the expert [11–13], or an imperfect expert [12], while in our work the deferral penalty (cognitive workload) is time-dependent.

To model the cognitive workload penalty, we considered a number of different reward models from the bandits literature that take into account the frequency of queries. Such approaches include explicit reward models that depend on the total number of queries [18, 20, 21] or the number of queries within a time interval [22], and methods that indirectly affect reward by shortening the time horizon [23]. We chose to apply the rebounding bandit model [18] to our contextual bandit setting due to its simplicity and expected qualitative similarity to real-world cognitive workload dynamics [24, 25].

**Shared autonomy for human-robot interaction.** A number of papers explore the concept of shared autonomy between a robot and human, in which the policies of the robot and human are combined in such a way that both contribute to the decision-making for a particular task. Shared autonomy approaches differ in the types of information that they infer about the human, such as the goals of the human [15, 16], and also in the degree to which human input guides the autonomous system [26]. Our work differs in that our algorithm decides between the decision of the agent and human input without blending the two, and we incorporate a model of human state (cognitive workload) distinct from the intent of the human.

# B   Hyperparameter Tuning

We describe our hyperparameter tuning process for the LinUCB baseline. Because all querying algorithms default to LinUCB as their action-selection method, we first select the LinUCB hyperparameters using our tuning process, and then use these fixed LinUCB hyperparameter values in our evaluations for the querying algorithms.

For hyperparameter tuning, in addition to the pretrain setting that we considered in our main experiments (shown in Figure 1), we considered the performance of LinUCB in three additional pretrain settings, each containing food items with different optimal robot actions. The four pretrain settings that we considered are: (1) no pretraining, (2) pretraining with food items whose optimal pitch angle is tilted vertical (TV), (3) pretraining with food items whose optimal pitch angle is either tilted angled (TA) or tilted vertical (TV), and (4) pretraining with food items whose optimal pitch angle is either tilted angled (TA) or vertical skewer (VS).

For LinUCB, we tune the $\alpha$ parameter, and we keep the $L_2$ regularization parameter $\lambda = 0.1$ fixed. We consider 9 distinct values of $\alpha$, ranging from 0.001 to 0.5, and we measured the mean convergence rate for the LinUCB policy across 5 random seeds. Each random seed controls the sequence of contexts that were observed as well as the stochasticity in the one-step rewards. Figure 3 shows the mean convergence rate across different values of $\alpha$, and across different generalization settings. We ultimately selected $\alpha = 0.1$ because it had the highest convergence rates across the four generalization settings with different choices for $D_p$ and $D_v$. Our hyperparameter tuning scheme is similar to that of prior empirical work for contextual bandits that leverages supervised datasets [27].

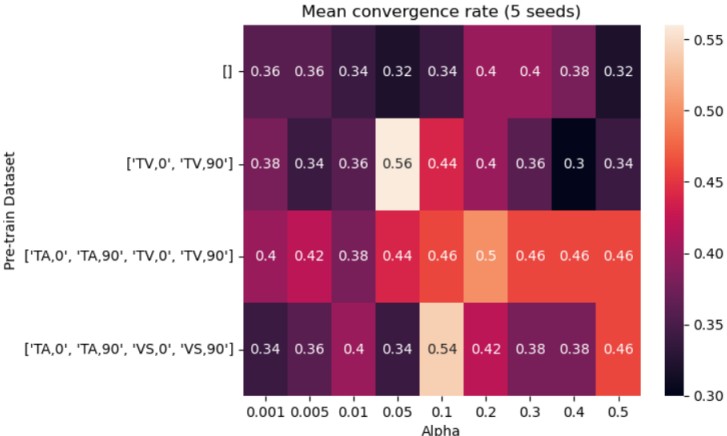

Figure 3: Hyperparameter tuning for LinUCB, showing mean convergence for different pre-train settings and different values of $\alpha$.

## C   Additional Experiments

**Fraction of querying statistics.** In Table 2, we augment the algorithm statistics shown in Table 1 to include the $f_q$ metric, indicating the fraction of timesteps in which each algorithm selects the querying action. The results show that LinUCB-QG achieves higher task performance through more frequent querying.

| **Method** | $f_q$ | $r_{task}$ | $r_{CL}$ | $M_{wt}$ |
|---|---|---|---|---|
| LinUCB | - | 0.716 | - | 0.501 |
| Exp-Query | 0.027 | 0.738 | $-\mathbf{0.026}$ | 0.509 |
| LinUCB-QG | 0.391 | **0.819** | $-0.139$ | **0.532** |

Table 2: Querying bandit algorithm metrics. Averages are across 5 random seeds (For Exp-Query, we also average across 5 policy random seeds). Values for $r_{task}$ and $r_{CL}$ correspond to hyperparameter setting with maximal $M_{wt}$.

**Results for convergence metric.** We also considered an additional set of metrics to capture task performance: *convergence*, where we define convergence $r_\kappa = \sum_{i=1}^{N_{fooditem}} \kappa_i$, where $N_{fooditem}$ is the total number of food items encountered in an episode, and $\kappa_i$ is a binary variable that is 1 if the algorithm selects the action with the highest ground-truth probability of success within 10 trials, and 0 otherwise, and *weighted-convergence*, given by $M_{wc} = w_{task}r_\kappa + w_{CL}r_{CL}(x, a, t)$. Table 3 and Figure 4 show the metric results and Pareto plot for the convergence metrics, respectively. The advantages in the weighted metric and task score are more pronounced when considering convergence compared to task reward in Figure 2, which we attribute to the discrete nature of the convergence metric, but LinUCB-QG still has superior weighted task reward compared to the other algorithms.

| **Method** | $r_\kappa$ | $r_{CL}$ | $M_{wc}$ |
|---|---|---|---|
| LinUCB | 0.4 | - | 0.28 |
| Exp-Query | 0.796 | $-0.072$ | 0.536 |
| LinUCB-QG | **0.98** | $-0.133$ | **0.646** |

Table 3: Querying bandit algorithm metrics. Averages are across 5 random seeds (For Exp-Query, we also average across 5 policy random seeds). Values for $r_{task}$ and $r_{CL}$ correspond to hyperparameter setting with maximal $M_{wc}$.

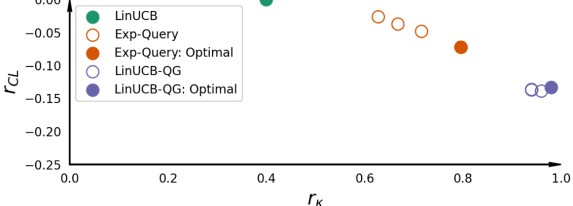

Figure 4: Pareto plot with $r_\kappa$ and $r_{CL}$ for LinUCB and different settings of Exp-Query (varying $c$) and LinUCB-QG (varying $w$). Filled circles indicate the setting that maximizes $M_{wc}$.

**Additional user settings.** In order to understand the effect that variable user settings for $w_{task}$ and $w_{CL}$ have on the relative performance of the querying algorithms, we experimented with three additional settings for these weight parameters: (1) $w_{CL} = 1.5w_{task}$, (2) $w_{CL} = w_{task}$, (3) $w_{task} = 1.5w_{CL}$, which correspond to values of $w_{task} = [0.4, 0.5, 0.6]$, respectively. Results for the weighted task and convergence metrics are shown in Tables 4 and 5, respectively.

We see that for smaller values of $w_{task}$ (larger values of $w_{CL}$), the cognitive workload of LinUCB-QG is amplified in the weighted metrics, leading it to have a smaller weighted score compared to LinUCB in Table 4. However, as $w_{task}$ increases, the improved task metric (convergence or task reward) for LinUCB-QG begins to dominate the weighted metric.

Table 4: Results for bandit algorithms for querying, using *weighted-task* metric. Averages are across 5 random seeds. For Exp-Query, we also average across 5 independent policy random seeds. Convergence and cognitive workload values for each algorithm correspond to hyperparameter setting with maximal weighted reward.

| $\mathbf{w_{task}}$ | **Method** | $r_{task}$ | $r_{CL}$ | $M_{wt}$ |
|---|---|---|---|---|
| | LinUCB | 0.716 | - | **0.286** |
| 0.4 | Exp-Query | 0.738 | $-\mathbf{0.026}$ | 0.279 |
| | LinUCB-QG | **0.819** | $-0.139$ | 0.244 |
| | LinUCB | 0.716 | - | **0.358** |
| 0.5 | Exp-Query | 0.738 | $-\mathbf{0.026}$ | 0.356 |
| | LinUCB-QG | **0.819** | $-0.139$ | 0.340 |
| | LinUCB | 0.716 | - | 0.430 |
| 0.6 | Exp-Query | 0.738 | $-\mathbf{0.026}$ | 0.432 |
| | LinUCB-QG | **0.819** | $-0.139$ | **0.436** |

Table 5: Results for bandit algorithms for querying, using *weighted-convergence* metric. Averages are across 5 random seeds. For Exp-Query, we also average across 5 independent policy random seeds. Convergence and cognitive workload values for each algorithm correspond to hyperparameter setting with maximal weighted reward.

| $\mathbf{w_{task}}$ | **Method** | $r_{\kappa}$ | $r_{CL}$ | $M_{wc}$ |
|---|---|---|---|---|
| | LinUCB | 0.4 | - | 0.16 |
| 0.4 | Exp-Query | 0.796 | $-\mathbf{0.072}$ | 0.275 |
| | LinUCB-QG | **0.98** | $-0.133$ | **0.312** |
| | LinUCB | 0.4 | - | 0.2 |
| 0.5 | Exp-Query | 0.796 | $-\mathbf{0.072}$ | 0.362 |
| | LinUCB-QG | **0.98** | $-0.133$ | **0.423** |
| | LinUCB | 0.4 | - | 0.24 |
| 0.6 | Exp-Query | 0.796 | $-\mathbf{0.072}$ | 0.449 |
| | LinUCB-QG | **0.98** | $-0.133$ | **0.535** |

## C.1 Results with time-delayed querying model.

We also considered a cognitive workload model that was time-delayed, where the value $c_{CL}(t)$ was delayed by one timestep (that is, the decision whether to query only affected the cognitive workload at the next timestep). This affects both the decision whether to query, as well as the overall cognitive workload reward $r_{CL}$ induced by querying. This setting is more similar to the satiation setting explored in [18], in that a time-delayed cognitive workload score is more lenient towards a single query but penalizes repeated querying after the initial query. In the below experiments, we used $\gamma = \gamma_q = 0.5$, corresponding to a higher one-step penalty of querying compared to the results in the main paper.

**Results for task and convergence metrics.** Results for the task and convergence metrics are shown in Tables 6 and 7, with $w_{task} = 0.7$ and $w_{CL} = 0.3$. We observe that in Figure 6 tuning the decay parameter of Exp-Query leads to a direct tradeoff between convergence and cognitive workload, but even with this tuning, its convergence is suboptimal compared to LinUCB-QG. Again, the advantages in the weighted metric and task score are more pronounced when considering convergence compared to task reward in Figure 5.

Compared to the cognitive workload setting in the main paper, we observe that LinUCB-QG achieves lower absolute cognitive workload and lower relative cognitive workload compared to Exp-Query, suggesting that LinUCB-QG efficiently asks queries leading to increased task performance.

| Method | $r_{task}$ | $r_{CL}$ | $M_{wt}$ |
|---|---|---|---|
| LinUCB | 0.728 | - | 0.510 |
| Exp-Query | 0.776 | $-0.625$ | 0.356 |
| LinUCB-QG | **0.807** | $-0.145$ | **0.521** |

Table 6: Querying bandit algorithm metrics. Averages are across 5 random seeds (For Exp-Query, we also average across 5 policy random seeds). Values for $r_{task}$ and $r_{CL}$ correspond to hyperparameter setting with maximal $M_{wt}$.

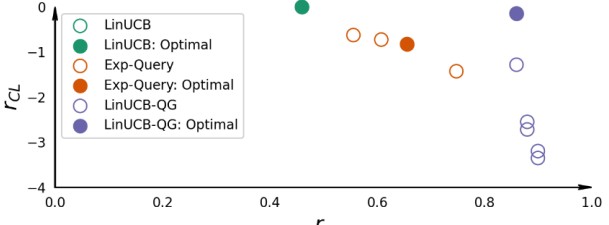

Figure 5: Pareto plot with $r_{task}$ and $r_{CL}$ for Lin-UCB and different settings of Exp-Query (varying $c$) and LinUCB-QG (varying $w$). Filled circles indicate the setting that maximizes $M_{wt}$.

| Method | $r_\kappa$ | $r_{CL}$ | $M_{wc}$ |
|---|---|---|---|
| LinUCB | 0.46 | - | 0.322 |
| Exp-Query | 0.656 | $-0.825$ | 0.212 |
| LinUCB-QG | **0.86** | $-0.145$ | **0.558** |

Table 7: Querying bandit algorithm metrics. Averages are across 5 random seeds (For Exp-Query, we also average across 5 policy random seeds). Values for $r_{task}$ and $r_{CL}$ correspond to hyperparameter setting with maximal $M_{wc}$.

Figure 6: Pareto plot with $r_\kappa$ and $r_{CL}$ for Lin-UCB and different settings of Exp-Query (varying $c$) and LinUCB-QG (varying $w$). Filled circles indicate the setting that maximizes $M_{wc}$.

**Additional user settings.** Tables 8 and 9 show values for the *weighted-task* and *weighted-convergence* metrics, respectively. Similar to the cognitive workload setting in the main paper, we see that smaller values of $w_{task}$ amplify the cognitive workload of LinUCB-QG, decreasing its weighted score compared to LinUCB in Table 8. However, as $w_{task}$ increases, the improved task metric (convergence or task reward) for LinUCB-QG begins to dominate the weighted metric.

Table 8: Results for bandit algorithms for querying, using *weighted-task* metric. Averages are across 5 random seeds. For Exp-Query, we also average across 5 independent policy random seeds. Convergence and cognitive workload values for each algorithm correspond to hyperparameter setting with maximal weighted reward.

| $\mathbf{w_{task}}$ | **Method** | $r_{task}$ | $r_{CL}$ | $M_{wt}$ |
|---|---|---|---|---|
| | LinUCB | 0.728 | - | **0.291** |
| 0.4 | Exp-Query | 0.776 | $-0.625$ | $-0.0646$ |
| | LinUCB-QG | **0.807** | $-0.145$ | 0.236 |
| | LinUCB | 0.728 | - | **0.364** |
| 0.5 | Exp-Query | 0.776 | $-0.625$ | 0.0754 |
| | LinUCB-QG | **0.807** | $-0.145$ | 0.331 |
| | LinUCB | 0.728 | - | **0.437** |
| 0.6 | Exp-Query | 0.776 | $-0.625$ | 0.216 |
| | LinUCB-QG | **0.807** | $-0.145$ | 0.426 |

Table 9: Results for bandit algorithms for querying, using *weighted-convergence* metric. Averages are across 5 random seeds. For Exp-Query, we also average across 5 independent policy random seeds. Convergence and cognitive workload values for each algorithm correspond to hyperparameter setting with maximal weighted reward.

| $\mathbf{w_{task}}$ | **Method** | $r_{\kappa}$ | $r_{CL}$ | $M_{wc}$ |
|---|---|---|---|---|
| | LinUCB | 0.46 | - | 0.184 |
| 0.4 | Exp-Query | 0.556 | $-0.625$ | $-0.153$ |
| | LinUCB-QG | **0.86** | $-0.145$ | **0.257** |
| | LinUCB | 0.46 | - | 0.23 |
| 0.5 | Exp-Query | 0.556 | $-0.625$ | $-0.0345$ |
| | LinUCB-QG | **0.86** | $-0.145$ | **0.357** |
| | LinUCB | 0.46 | - | 0.276 |
| 0.6 | Exp-Query | 0.556 | $-0.625$ | 0.0836 |
| | LinUCB-QG | **0.86** | $-0.145$ | **0.458** |

