# OpenReview forum: "To Ask or Not To Ask: Robot-assisted Bite Acquisition with Human-in-the-loop Contextual Bandits"
_robot-learning.org/CoRL/2023/Workshop/OOD — OOD Workshop @ CoRL 2023_

### Official Review · Reviewer_KEFR · 2023-10-16
**Writing needs to be a bit more accessible to non-experts in contextual bandits**

**Rating:** 6
**Confidence:** 2

**Review:**

This paper proposes a contextual bandit algorithm to a robot-assisted bite acquisition task. The algorithm asks for human feedback to improve OOD generalization to novel food items, while minimizing the cognitive workload that querying imposes on the human.

The results in the paper look interesting and relevant to the workshop. But I would recommend the authors to update the paper to make it more accessible to non-experts, like me, in contextual bandits. I did not follow the main technical contribution of the paper. It would be good to highlight the key high-level insight behind the technical contribution for non-expert readers.

---

### Official Review · Reviewer_ogtT · 2023-10-16
**Online Methodology for OOD Generalization**

**Rating:** 6
**Confidence:** 3

**Review:**

This paper presents an online methodology using contextual multi-arm bandits (cMABs) to generalize a bite acquisition task. The paper proposes utilizing a cognitive load cost as a regularizer for seeking expert assistance, in order to balance correctness (good 'grasps') with effort for an 'expert' labeler. Though not considering 'traditional' OOD notions, the results provided utilize online methods that in principle provide regret bounds, which hold uniformly for all realizations of uncertainty and thus for OOD instances.

General comments / thoughts:
- Paper is generally clearly written but the exposition is confusing or surprising at times. As one example: the action space has cardinality six, but it's actually seven if we include the querying action, but in some sense just four when we reduce everything to three broader categories and the query action, etc. After a while, I was able to settle on "seven actions, arranged in the (3x2 + 1)" format, but it took a little while to get there! Introducing the three acronyms in the action space discussion might help clarify this.
- Similarly, the feature/context space seems impossibly large at first sight, because it runs against the 'usual' grain of MAB and cMAB methods; such a large feature space would seem to have to preclude meaningful regret guarantees (except possibly under the assumption of realizability of the linear regression?).
- There are a few frequentist statistics that might be useful to include: (a) fraction of query actions (the reward [CL] function in Table 1 does not seem to be easily invertible as to allow the reader to reconstruct this), and (b) three-category contextual optimum mean rewards -- that is, some kind of $\bar{r}(x, a^*)$ for all x in (resp) contextual group TA (TV, VS). These three numbers would be useful in calibrating the convergence (in practice) to the optimum solution.
- The regularizer QG in the proposed algorithm is an interesting idea that might extend to a variety of other HRI contexts.
- Some additional inquiry into the Pareto frontier would be an interesting addition, that might also allow for understanding calibration of query rate for different mean rewards.

---

### Decision · Program_Chairs · 2023-10-17

**Decision:**

Accept

**Comment:**

We agree with the reviewers’ assessment that this work is technically sound and will contribute to productive, topical discussions at the 2023 Workshop on OOD Generalization in Robotics. In particular, we appreciate that the problem setup of this work explicitly considers OOD generalization performance, though the clarity of exposition for the method could be improved. We recommend the authors incorporate the reviewers’ feedback into their camera-ready submission to further improve their manuscript.